# Measurement of 17-Hydroxyprogesterone by LCMSMS Improves Newborn Screening for CAH Due to 21-Hydroxylase Deficiency in New Zealand

**DOI:** 10.3390/ijns6010006

**Published:** 2020-01-28

**Authors:** Mark R. de Hora, Natasha L. Heather, Tejal Patel, Lauren G. Bresnahan, Dianne Webster, Paul L. Hofman

**Affiliations:** 1Newborn Screening, Specialist Chemical Pathology, LabPlus, Auckland City Hospital, Auckland 1023, New Zealand; NHeather@adhb.govt.nz (N.L.H.); tejalP@adhb.govt.nz (T.P.); lbresnahan@adhb.govt.nz (L.G.B.); diannew@adhb.govt.nz (D.W.); 2Clinical Research Unit, Liggins Institute, University of Auckland, Auckland 1010, New Zealand; p.hofman@auckland.ac.nz

**Keywords:** congenital adrenal hyperplasia, 17-hydroxyprogesterone, newborn screening, liquid chromatography tandem mass spectrometry

## Abstract

The positive predictive value of newborn screening for congenital adrenal hyperplasia due to 21-hydroxylase deficiency was <2% in New Zealand. This is despite a bloodspot second-tier immunoassay method for 17-hydroxyprogesterone measurement with an additional solvent extract step to reduce the number of false positive screening tests. We developed a liquid chromatography tandem mass spectrometry (LCMSMS) method to measure 17-hydroxyprogesterone in bloodspots to replace our current second-tier immunoassay method. The method was assessed using reference material and residual samples with a positive newborn screening result. Correlation with the second-tier immunoassay was determined and the method was implemented. Newborn screening performance was assessed by comparing screening metrics 2 years before and 2 years after LCMSMS implementation. Screening data analysis demonstrated the number of false positive screening tests was reduced from 172 to 40 in the 2 years after LCMSMS implementation. The positive predictive value of screening significantly increased from 1.71% to 11.1% (X^2^ test, *p* < 0.0001). LCMSMS analysis of 17OHP as a second-tier test significantly improves screening specificity for CAH due to 21-hydroxylase deficiency in New Zealand.

## 1. Introduction

Congenital adrenal hyperplasia (CAH) represents a group of inherited disorders characterised by absent or reduced adrenal cortisol synthesis. Approximately 90% of CAH is caused by mutations in the CYP21A2 gene resulting in reduced activity of adrenal steroid 21-hydroxylase. 21-Hydroxylase catalyses the conversion of 17α-hydroxyprogesterone (17OHP) to 11-deoxycortisol and progesterone to 11-deoxycorticosterone, the respective precursors to cortisol and aldosterone [1].

Reduced synthesis of aldosterone can lead to life threatening salt wasting, vascular collapse and an Addisonian crisis in the neonatal period. Reduced cortisol synthesis results in a loss of normal feedback to the hypothalamus and pituitary gland leading to an increase in pituitary adrenocorticotrophic hormone (ACTH) release. Increased ACTH levels cause adrenal hyperplasia with increased androgen synthesis and the concomitant rise in intermediate metabolites in the steroidogenesis pathway.

Many countries now perform newborn screening for CAH due to 21-hydroxylase deficiency to prevent life threatening salt-wasting crises and hypoglycaemia in early infancy. The screening test involves measurement of 17OHP by immunoassay in dried blood collected onto specialised blood collection paper [2]. Screening is sensitive for the severe salt-wasting form of CAH but can be confounded by high concentrations of cross reactive 17OHP steroid precursors and their sulphated conjugates, which are present in the first 48 h after birth and longer in pre-term neonates [3]. 17OHP levels may also be elevated due to illness, stress and biological variation [4,5].

In New Zealand, newborn screening specimens with an elevated 17OHP by immunoassay are subjected to a second-tier immunoassay after solvent extraction to remove polar steroids conjugates.

The goal of second-tier testing is to confirm elevated 17OHP levels and thus reduce the number of false positive screening results. The number of falsely elevated results however, remains high as non-polar interfering steroids, such as pregnenolone and 17OH-pregnenolone are not removed by solvent extraction [6]. In 2016, the positive predictive value of CAH screening in New Zealand was estimated at 1.08% [7] meaning that 100 neonates have unnecessary specimen recollections for every case of CAH detected.

Liquid chromatography tandem mass spectrometry (LCMSMS) offers a more analytically specific method of 17OHP measurement as compared with immunoassay. LCMSMS is now sufficiently sensitive for steroid analysis in bloodspots but chromatography is required to separate isobaric steroids and reduce matrix ion suppression. The technique is not suitable for high throughput population screening but is routinely used by screening programmes as a second tier test to confirm positive newborn screening tests or reduce the number of false positives for a range of metabolic disorders [8].

Our goal was to develop a simple and reliable method to measure 17OHP in bloodspots as a second-tier test to reduce the false positive rate of newborn screening for CAH in New Zealand. Correlation between LCMSMS and the immunoassay for 17OHP is reported to be poor in neonates [9] but better in 17OHP bloodspot reference material [10]. Therefore, our approach was to develop a LCMSMS method and determine its performance using 17OHP bloodspot reference material, proficiency testing samples from a quality assurance scheme and bloodspots enriched with 17OHP prepared in our laboratory. We determined the relationship between LCMSMS and immunoassay in 17OHP proficiency testing samples from a certified quality assurance programme and in bloodspots from neonates with false positive and true positive newborn screening results to confirm the suitability of LCMSMS for our screening protocols. We then implemented LCMSMS as a second-tier test and assessed the performance by comparing screening parameters prior to and after LCMSMS implementation.

## 2. Materials and Methods

### 2.1. Reagents

17-hydroxyprogesterone and formic acid were purchased from Sigma-Aldrich (Auckland, NZ), d8-hydroxyprogesterone was purchased from SCIVAC (Hornsby, NSW, Australia). 17-hydroxyprenenolone sulphate was purchased from Steraloids (Newport, Rhode Island, USA). LCMSMS grade acetonitrile, methanol, acetone and isopropanol were purchased from Thermo Fisher NZ (Auckland, NZ).

### 2.2. Calibrators and Controls

Calibrators and control materials were manufactured using donor whole blood with plasma run off (New Zealand Blood Service). Red cells were washed 3 times with saline before 55% haematocrit blood was made using a serum substitute (Serasub, CST Technologies, NY, USA) to replicate neonatal blood. A stock standard steroid solution was made by dissolving 33.1 mg 17OHP in 10 mL ethanol (10 mM 17OHP) and diluting in saline to make a 33.33 µM spiking solution. Bloodspot 17OHP calibrators were made by adding 0, 15 µL, 30 µL, 60 µL, 120 µL and 240 µL of spiking solution to 20 mL aliquots of 55% haematocrit blood to make 6 calibrators ranging from 0 to 400 nmol/L. Three controls were prepared using the same procedure with 10 µL, 45 µL and 90 µL of 17OHP spiking solution. Blood was mixed for 1 h and spotted onto blood collection paper (Whatman 903™), dried at room temperature and stored at −20 °C until analysis.

A stock solution of internal standard was prepared by dissolving 1.25 mg of d_8_-17OHP in 500 mL methanol divided into glass vials and stored at −80 °C. A working internal standard (36.9 nmol/L) was prepared by diluting 5 mL stock in 50 mL methanol.

To evaluate the interference of sulphated steroids, 17-hydroxypregnenolone sulphate enriched samples were prepared by adding 240 µL, 120 µL, 60 µL and 30 µL of 333.3 µM solution to 20 mL of 55% haematocrit blood. Blood was mixed for 1 h and spotted onto blood collection paper, dried at room temperature and stored at −20 °C until analysis. The final concentrations of 17-hydroxypregnenolone sulphate were 500, 1000, 2000 and 4000 nmol/L.

### 2.3. Bloodspot Samples

148 residual newborn screening specimens with out-of-range 17OHP levels (previously used for second-tier immunoassay) were subjected to LCMSMS. Of these, 132 were from infants without CAH (i.e., they had a false positive screen result) and 16 from affected infants.

Certified quality control material (3 levels) enriched with known quantities of 17OHP were used to assess accuracy and recovery. Certified material was supplied by the Centres for Disease Control and Prevention (CDC, Atlanta, USA) Newborn Screening Quality Assurance Scheme.

Control material (3 levels) prepared with the method calibrators were used to assess precision. The interference from sulphated steroid conjugates was assessed using 17-hydroxypregnenolone sulphate enriched bloodspot material.

Thirty three residual external quality assurance samples (EQA) supplied by a CDC Newborn Screening Proficiency Scheme were used to determine the correlation between LCMSMS and immunoassay methods of analysis.

### 2.4. Immunoassay Protocols of CAH Newborn Screening

Newborn screening specimens are collected from neonates by heel prick onto blood collection cards 48–72 h after birth. In New Zealand, further samples are collected at 2 weeks for babies born with a birthweight (BW) of ≤1500 g and a third sample is collected at 4 weeks if the BW < 1000 g. A screening test is considered positive if a further unscheduled intervention, such as a sample recollect or a clinical referral is warranted.

The primary screening test for CAH due to 21-hydroxylase deficiency is 17OHP measurement carried out using a time resolved fluoroimmunometric direct immunoassay (Perkin Elmer, Turku, Finland). Newborn screening specimens were subjected to an additional immunoassay analysis after solvent extraction when the primary test result was above the newborn screening cut-off. For the extracted immunoassay, a single 3 mm bloodspot, calibrators and controls were punched in 1.5 mL polypropylene tubes. Bloodspots were eluted with 200 µL of 0.1 M phosphate buffer followed by extraction with 1 mL of diethyl ether. Extracts were transferred to glass tubes, dried under nitrogen, reconstituted in immunoassay kit buffer and analysed. Bloodspot calibration material (6 levels) is included in kits by the manufacturer with ranges from approximately 10 nmol/L to 300 nmol/L.

For the primary screening test, 17OHP concentrations by direct immunoassay of ≥37 nmol/L in neonates with a birth-weight (BW) ≤1500 g or ≥27 nmol/L if BW >1500 g were considered out of range and subjected to a second-tier test. The second-tier immunoassay test was considered out of range and screen positive for CAH if the 17OHP concentration was ≥34 nmol/L in neonates with a BW ≤ 1500 g or ≥24 nmol/L in neonates with a BW > 1500 g. A CAH screen is considered positive if further action is required on the baby, i.e., and additional sample or a clinical referral. Hence, an out-of-range result at 48 h on a 1400 g BW baby would not be considered a positive screen as a further sample is scheduled to be taken at 2 weeks.

### 2.5. Sample Preparation for LCMSMS Method

For the LCMSMS method, 2 × 3 mm (6.4 µL) dried blood spots were added to a 96-deep well polypropylene micro titre plate. 20 µL of internal standard was added to each well followed by 200 µL of 80% acetonitrile in water. Plates were sealed and mixed gently for 1 h on a plate mixer and spun at 3000 rpm for 5 min. 200 µL of supernatant was transferred to a 96-shallow well micro titre plate, dried under nitrogen at 50 °C and reconstituted with 80 µL of 40% methanol in water. Plates were covered in foil and mixed gently for 10 min on a plate mixer before analysis.

### 2.6. LCMSMS Analysis

The LCMSMS system comprised an ultra high pressure liquid chromatography (UHPLC) quaternary pump combined with a TSQ Vantage mass spectrometer both from Thermo Fisher (Waltham, Mass., USA). An in-line Thermo Fisher turbulent flow (TLX) solid phase extraction was available for use. It comprised of a second UHPLC pump and a cyclone-P Turboflow™ (0.5 × 50 mm) extraction column. TLX technology works as follows. Prepared samples are injected onto the turboflow column at high flow rate using the first UHPLC pump (Loading pump). A combination of diffusion and size exclusion results in the retention of low molecular weight (<600 amu) polar and non-polar compounds. High molecular weight compounds are not retained and flow to waste. The turboflow column is then eluted with solvent and sample is transferred to an analytical column for chromatography separation and detection by mass spectrometry. The method removes protein particulates and phospholipids which can have a negative impact on analytical chromatography column performance and lifetime.

Solvent A consisted of ultra-pure water (Milliq, Merck, Darmstat, Germany) with 0.05% formic acid and 2 mM ammonium formate. Solvent B was 100% methanol. Solvent C was 100% acetonitrile. Solvent D was acetonitrile/isopropanol/acetone (45:45:10).

The turboflow UHPLC method contains a series of steps that control pump flow rate, valve positions, step duration and mobile phase composition. Briefly, the prepared samples were injected onto the turboflow column, via a 20 µL sample loop, at a flow rate of 1.5 mL/min (95% solvent A and 5% solvent C). Bound material was transferred to the analytical column by eluting the turboflow column with 80% acetonitrile from an inline elution loop at a low flow rate (0.1 mL/min) for 1 min. Chromatographic separation was achieved by ramping of mobile phase B to 90% in 5 min before holding for 1 min at 0.5 mL/min. During the chromatography phase the turboflow column was washed with 100% mobile phase D before the elution loop was refilled with 80% acetonitrile. The total run time was 9.15 min. Detailed chromatography settings are shown in Table A1.

The capillary voltage on the ion source was 3000 volts, the capillary temperature was 320 °C, the vaporiser temperature was 450 °C, sheath gas pressure was 50, ion sweep pressure 1.0 and auxiliary gas pressure 15 (arbitrary units). The quantifier precursor and product ions were 331.3 > 97.1 for 17 OHP and 339.3 > 100.1 for 17OHP-d_8_. The qualifier ion transition for 17OHP was 331.3 > 109.1. The scan width was 0.050 m/z for each transition with a scan time of 0.020 s. The de-clustering voltage was 3 V.

### 2.7. LCMSMS Quantification

Data processing was performed using Tracefinder software provided with the instrument. Peak area ratios of 17OHP/17OHP-d_8_ were plotted against standard concentrations to achieve a linear calibration curve plotted with a 1/X reciprocal fit weighting to ensure maximum accuracy at lower concentrations.

### 2.8. LCMSMS Assay Performance Assessment

Accuracy and recovery were assessed using reference material with known 17OHP enrichment supplied by the CDC. Recovery was calculated from the baseline 17OHP values, enriched values supplied with the samples and the LCMSMS results in 3 concentration levels. Within and between batch precision was assessed in 3 levels of control material prepared in the laboratory. Linearity was assessed using incremental quantities of calibration material. The influence of ion suppression was assessed by monitoring of a post column infusion of 17OHP-d_8_ in 5 extracted bloodspots using a previously described procedure [11]. The lower limit of quantification (LOQ) was assessed by repeated analysis of enriched bloodspot samples with 17OHP concentrations of 1–10 nmol/L.

Newborn screening performance incorporating LCMSMS was determined by reviewing CAH newborn screening data (*n* = 117,063 neonates) 2 years after LCMSMS implementation. Data was compared to screening performance in newborns (*n* = 116,097) born in the 2 years prior to LCMSMS implementation.

### 2.9. Statistical Analysis and Comparison between LCMSMS and Immunoassay

The relationship between LCMSMS and immunoassay after solvent extraction was determined using methods described by Bland and Altman [12]. Difference plots were constructed using the mean percentage difference between LCMSMS and immunoassay for each specimen and the mean of both methods. The average difference and 2 standard deviation ranges for the limits of agreement were calculated.

The correlation between LCMSMS and immunoassay was determined in EQA proficiency material. Residual EQA specimens (*n* = 30) with concentration ranges spanning the primary screening cut-offs were subjected to LCMSMS. Solvent extraction was not carried out on these specimens because they were insufficient for second tier immunoassay measurement and they would not be expected to contain immunoassay interfering compounds.

The false positive rate, the sensitivity, specificity and positive predictive value of screening before and after LCMSMS implementation was determined. The Pearson’s Chi squared test (X^2^) was used to determine the significance of screening outcome and choice of second-tier method.

## 3. Results

### 3.1. LCMSMS Method Performance

Full chromatographic baseline separation for 17OHP from other potentially interfering isobaric compounds was achieved (Figure 1). Monitoring of the 17OHP internal standard response (m/z 339.25 > 100.2) in five extracted samples revealed no drop in internal standard response between 6.5 and 7.5 min (the elution time of 17OHP), an indication that ion suppression had little effect on 17OHP quantitation.

The average slope of 10 calibration curves carried out on separate days was 0.006711 (sd 0.000486) with a CV of 7.2%. Linearity was established up to 1600 nmol/L (*r*^2^ = 0.999, Figure 2).

The average within batch precision was 9.1% (*n* = 20) with between batch precision of 9.1% (*n* = 22) across 3 levels of control material. Accuracy was within an average +2 nmol/L of 10 replicates of reference material with target ranges of 17 nmol/L, an average of −5 nmol/L with a target range of 82 nmol/L and an average of +2 nmol/L for a target range of 162 nmol/L (Table 1).

The average recovery for the method was 100.5% with a range of 98.6–102% at three bloodspot concentrations of 17OHP. The lower limit of quantification (LOQ) was defined as the lowest concentration of 17OHP with an inter-batch precision of <20%. The LOQ for 17OHP was determined to be 1 nmol/L. Results from analysis of 17-hydroxypregnenolone sulphate enriched bloodspots revealed no interference with 17OHP measurements (Table 2).

### 3.2. Correlation between LCMSMS and Immunoassay

Bland Altman analysis to determine the difference between LCMSMS and immunoassay in false positive neonatal screening samples revealed a mean difference of −36.1% with 2 standard deviation limits of agreement of −97.8% to 26.4% (Figure 3). Analysis in the true positive CAH screening samples revealed a mean difference of −45.5% with limits of agreement of −90.7% to 0.4%. Correlation of LCMSMS and immunoassay in EQA samples revealed a difference of −11.1% with limits of agreement between 38.8% and 27.6% (Figure 4). The Bland Altman plot regression line slopes for each sample group was not significantly different from 0 (*p* > 0.05), indicating an insignificant change in bias across the measured range for each group.

### 3.3. Analysis of Newborn Screening Data

During the 2 years prior to LCMSMS implementation 1643 s tier immunoassay tests were performed. Of these 362 samples were above the newborn screening cut-off with 175 positive screening results. During that time there were three clinically proven cases of P450c21 deficiency and 172 false positive tests requiring follow up. In the 2 years after LCMSMS implementation there were 1213 s tier LCMSMS tests with 113 samples above the screening cut-off resulting in 45 positive screens. Of these 5 were clinically confirmed cases of CAH due 21-hydroxylase deficiency and 40 were considered false positive tests requiring follow up. The data indicated there was a significant relationship between the second-tier method of analysis and the number of false positive screening results using the same newborn screening 17OHP cut-offs. (Pearsons Chi-Squared statistic, X^2^ = 47.29, *p* < 0.00001, *n* = 2856). The positive predictive value of newborn screening for CAH due to 21-hydroxylase deficiency increased from 1.71% to 11.1% (Table 3) when LCMSMS was introduced. LCMSMS significantly improves screening specificity for CAH without any other changes to newborn screening protocols.

The birth weights, gestational age (GA) at birth, the age of sampling and 17OHP are shown in Table A2. The table also includes the corrected gestational age (GA at birth + age at sampling). The GA of babies at birth ranged from 23 to 36 weeks with corrected GA of 27–36 weeks. All false positive specimens were collected in hospital neonatal units or neonatal intensive care units. In total, 16 samples were from neonates screened under the low birth weight protocol. In total, 24 samples were from neonates screened under the single sample screening protocol. All neonates with a false positive screen were premature (<37 weeks gestational age).

## 4. Discussion

We have described a LCMSMS method to measure 17OHP in bloodspots as a second-tier test for newborn screening for CAH. The method was based on a procedure described by Rossi et al. [13] with an additional automated sample clean up step by turbulent flow solid phase extraction. Our inter assay method precision range of 9.3–11.9% was comparable to other reports (7.9–10.9% [10], 3.9–18% [14]). The method is also sufficiently accurate, particularly when using the inter laboratory method mean as a target for certified CDC material. The method was linear beyond the clinical requirements for CAH investigations. The mean recovery for three 17OHP concentrations was almost 100%, although a repeat recovery experiment in native blood to account for protein binding of 17OHP may be have been more appropriate. However, we assumed that protein precipitation would release bound 17OHP in the sample when the acetonitrile solution was added during the extraction phase of the procedure. There was no interference detected from sulphated steroids when added at concentrations expected in bloodspot specimens from very premature neonates. The limit of quantification was 1 nmol/L.

Salter et al. [9] and Janzen et al. [10] found poor correlation between LCMSMS and radioimmunoassay in a small number of CAH patient samples. When comparing the two techniques, Boelen et al. reported that LCMSMS measurements in neonatal samples were up to 75% lower than immunoassay [15], while Dhillon et al. showed that the difference between LCMSMS and immunoassay was much wider in very low birth weight babies [16]. Our data confirms their reports that LCMSMS 17OHP is significantly lower and the differences are highly variable, particularly in low birth weight babies.

The correlation between LCMSMS and immunoassay is improved when external quality assurance proficiency testing samples are used. LCMSMS is, on average, 11% lower than immunoassay. The reasons for this are unclear as the samples would not be expected to contain large quantities of cross reacting steroid precursor. There is evidence that sample disks taken from the outer edges of a bloodspot can have up to 25% lower concentrations of metabolites than sample disks taken from the centre of a bloodspot [17]. It is common practice in screening laboratories to punch disks from the centre of a bloodspot. During our method development the restricted quantities of residual EQA material and newborn screening specimens meant that disks where often taken from the outer edges of the bloodspot.

LCMSMS removes interference from other steroids but can be prone to interference from other isobaric steroids, a number of which can be present in newborn bloodspots. 11-deoxycorticosterone, an aldosterone precursor, has previously been shown to interfere with 17OHP measurements by LCMSMS [9] when present in large quantities. Although concentrations are normally much lower than 17OHP and even absent in 21-hydroxylase deficiency, we did not note any interfering peaks in our method in any of our specimens.

For newborn screening to be effective, positive tests need to be confirmed quickly so clinical referrals can be made. Our method allows LCMSMS to be carried out quickly after an initial elevated primary immunoassay test. Calibration stability suggests that periods between full method calibration only need to be performed every 10 days, therefore second-tier results will most often be available within 2 h when required.

This approach has significantly improved screening specificity for CAH in New Zealand. The number of false positive screening tests have fallen from 86 to 20 per year, a reduced burden on the screening programme and a reduction in screening interventions. False positive tests are expensive as they require follow up, sample recollection and analysis, and can cause lasting anxiety for families [18]. All of our false positive screening tests were from samples taken in specialist neonatal units in New Zealand hospitals. The risk of an undetected salt wasting crisis occurring in this group of babies is low, and a protocol that includes specimen collection at unit discharge, (i.e., at a corrected gestational age that is near-term) may reduce the false positive rate further.

The use of 17OHP and androstenedione expressed as a ratio to cortisol has been used as additional secondary marker in CAH [14,19,20]. In a report by Schwarz et al. on the use of LCMSMS incorporating 17OHP and a ratio of androstenedione plus 17OHP to cortisol (17OHP+A4/CORT) resulted in a significant reduction of the false positive rate in CAH screening from 2.64% to 0.09% over a 13 month period [20]. Using 17OHP alone, our data showed the false positive rate was reduced from 0.15% to 0.03% after the introduction of LCMSMS. The second tier 17OHP cut-off was higher in New Zealand (23–34 nmol/L) when compared to a 17OHP cut-off of 17.3 nmol/L (12.5 mg/mL serum equivalent). Additionally, the strategy for repeat sampling in low birth weight neonates in New Zealand may have contributed significantly to a lower false positive rate than a single sample approach at the same age (2–3 days). The protocols used by both programmes point to the broader issue of a lack of a common approach to CAH screening in general. Comparing screening performance is difficult when there is a lack of consensus across most CAH screening programmes for an approach to timing of sample collection, screening cut-offs, laboratory protocols, the definition of what constitutes a positive screening test and a definition of what form of CAH screening programmes are trying to detect in the newborn period.

There are however a number of additional analytical strategies that could be implemented to improve screening specificity further. The analysis of additional informative markers in CAH has been employed by measuring 17OHP with a combination of androstenedione, 11-deoxycortisol, 21-deoxycortisol, cortisol and several other steroids as a second tier test [10,13,14,15,16,19]. 21-deoxycortisol is emerging as a sensitive marker for CAH although it requires further evaluation [15]. Further evaluation of some of or all these markers to improve screening in New Zealand may be appropriate.

The National Newborn Metabolic Screening Programme in New Zealand is accredited under ISO15189 and is required to take part in external quality assessment. While a number of proficiency schemes for bloodspot 17OHP by immunoassay and LCMSMS are available worldwide, no regular scheme is available for bloodspots that include additional informative steroids outside of the United States. Our approach was to develop a LCMSMS method for 17OHP alone, evaluate its screening performance, while participating in regular external quality assessment using certified schemes. This has allowed the screening laboratory to gain technical experience of LCMSMS and to build a platform to add additional markers. Alternatively a small adjustment to the sampling protocols (e.g., a final sample collect at neonatal unit discharge) combined with a second tier LCMSMS 17OHP level may offer a more simple approach to optimising CAH screening specificity. Both of these strategies require further study.

Although LCMSMS analysis of 17OHP offers improved analytical specificity, the limited data from the true positive cases does not indicate a change in screening sensitivity. Screening in New Zealand targets the severe salt-wasting form of 21-hydroxylase deficiency, however neonates with the simple virilising form of the disorder are detected by the screening programme and benefit from early treatment. Some neonates with 21-hydroxylase deficiency in New Zealand have been missed by screening and presented later with the symptoms of androgen excess. Our method offers the potential to add additional markers to improve screening sensitivity. For example, 21-deoxycortisol has been shown to be a consistent marker for CAH in the neonatal period [15] and 11-deoxycortisol can be used to distinguish 21-hydroxylase deficiency from 11-hydroxylase deficiency which is present in approximately 5% of CAH cases [15]. A number of novel markers for CAH are emerging and there is considerable interest in role of 11-oxygenated 21-carbon steroids in CAH [21].

In summary, the replacement of immunoassay by LCMSMS as a second-tier test has significantly improved newborn screening specificity for CAH due to 21-hydroxylase deficiency in New Zealand. Small adjustments in a sampling protocols or additional steroid markers may contribute to improved specificity and sensitivity. Analysis of 17OHP in bloodspots is a simple and reliable procedure and results are available in the timeframes needed for clinical referral when required.

## Figures and Tables

**Figure 1 IJNS-06-00006-f001:**
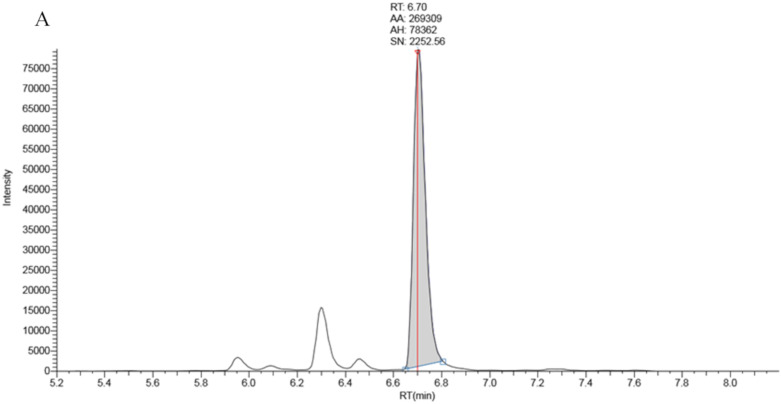
Chromatography traces for the 331.3 > 97.2 MRM signal by LCMSMS for **A**: True positive second tier LCMSMS screening test for CAH (17OHP = 401 nmol/L), **B**: Negative LCMSMS CAH screening test at day 2 on a baby born at 32 weeks gestation (17OHP = 7 nmol/L).

**Figure 2 IJNS-06-00006-f002:**
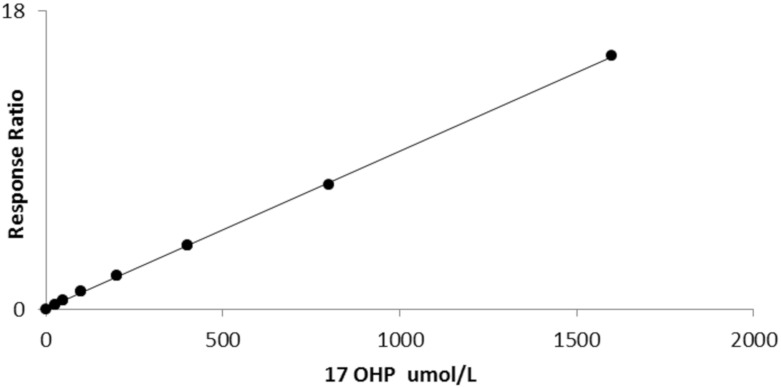
Linearity of bloodspot 17OHP by LCMSMS (*r*^2^ = 0.999).

**Figure 3 IJNS-06-00006-f003:**
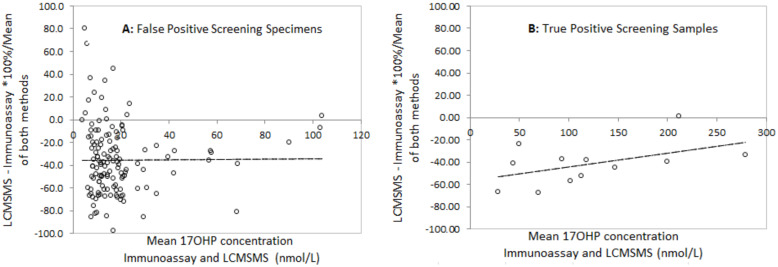
Bland Altman difference plots for LCMSMS and Immunoassay after solvent extraction for **A**: False positive screening Tests and **B**: True Positive Screening tests, and for LCMSMS and Immunoassay for External quality assurance samples. **A**: Slope = 0.02 (*p* = 0.89), mean difference = −35.7%, 2 sd limits of agreement (LOA) −97.9% to +26.4%); **B**: Slope = 0.13 (*p* = 0.15), mean difference = −45.5%, LOA= −90.7% to −0.4%).

**Figure 4 IJNS-06-00006-f004:**
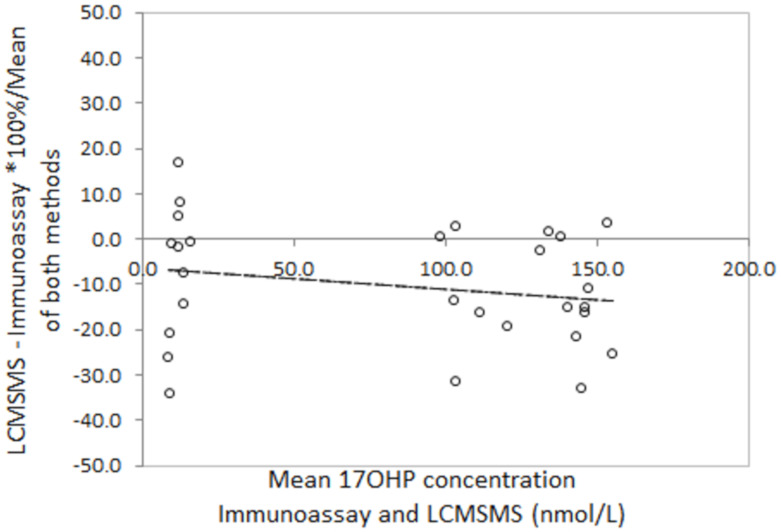
Bland Altman plots for LCMSMS and Immunoassay for external quality assurance samples (*n* = 32). Slope = −0.02 (*p* = 0.52), mean difference = −11.1%, LOA = −38.8 to 27.6%.

**Table 1 IJNS-06-00006-t001:** Precision, Accuracy, bias and recovery data for 17OHP in bloodspots by LC-MSMS. * Target value assigned by the CDC. ** LCMSMS method mean (*n* = 79) from laboratories submitting analytical data to the CDC.

Precision Data (*n* = 24)			
	Control 1	Control 2	Control 3
Within-Batch (*n* = 20)			
Mean (nmol/L)	14.6	74.9	155.7
CV%	13.7	7.0	6.2
Between-Batch (*n* = 22)			
Mean (nmol/L)	15.0	74.9	155.1
CV%	11.9	6.1	9.3
**Accuracy**			
	**Level 1**	**Level 2**	**Level 3**
CDC Target Value (nmol/L) *	15	66	134
Inter-laboratory Mean (nmol/L) **	17	82	162
Mean (nmol/L) (*n* = 10)	19	77	164
**Recovery**			
	**Level 1**	**Level 2**	**Level 3**
Baseline Level	ND	ND	ND
Enrichment (nmol/L)	15.2	76.0	152
Mean	15.0	74.9	155
Recovery (%)	101%	98.6%	102%

**Table 2 IJNS-06-00006-t002:** Comparison of interferences from 17-hydroxpregnenolone sulphate.

17OH-Pregnenolone Sulphate Conc. (nmol/L)	Immunoassay17OHP (nmol/L)	LCMSMS17OHP (nmol/L)
500	2.8	<1
1000	5.8	<1
2000	8.8	<1
4000	23.5	<1

**Table 3 IJNS-06-00006-t003:** Newborn Screening Metrics for CAH before and after LCMSMS Implementation.

	Immunoassay 2 nd Tier	LCMSMS 2 nd Tier
Start Date	1 December 2015	1 December 2017
End Date	30 November 2017	30 November 2019
Number of Babies Screened	117,063	116,097
Number of Specimens	118,624	117,624
Number of second tier tests	1643	1213
Number of 2 nd tier results above the screening cut-off	362	113
Number of positive screens	175	45
Number of true positive screens	3	5
Number of false positive screens	172	40
False Positive Rate	0.15%	0.03%
Specificity	99.85%	99.97%
Positive predictive Value	1.71%	11.11%

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
