# Peer review of "Measurement of 17-Hydroxyprogesterone by LCMSMS Improves Newborn Screening for CAH Due to 21-Hydroxylase Deficiency in New Zealand"

_2409-515X, 2020, doi:10.3390/ijns6010006_

Round 1

Reviewer 1 Report

There is only minor suggestion to improve techical style of references.

Author Response

Thank you for your review. I will review the technical style of the references in a re-submitted manuscript.

Reviewer 2 Report

This well written paper of de Hora et al provides a summary of local New Zealand experience with CAH screening using a 2-tiered immunoassay and LC-MS/MS approach. This approach is well established and widely used within the newborn screening community. However, the data presented by the authors may be of interest as it is focused on 17-OHP as sole marker and argues against the widely used algorithm which includes analytes such as androstenedione and cortisol as additional markers. This paper is suitable for publication once the following minor points are adequately addressed:

The number of positive screens after the use of LC-MS/MS is confusing. On line 247, the authors indicate 45 but the total of true positives (5) and false positives (35) is 40. In Table 3, the “number of positive screens” with LC-MS/MS as second tier was 40. In Appendix A, the authors listed 40 false positives (as opposed to 35 mentioned in the text and in Table 3). Clarification of this is required. In Table 3, the PPV with second tier immunoassay is incorrect. The correct PPV is 1.71. Please check. Provide additional table summarizing the levels of 17OHP in the 1 st tier screening immunoassay and that by LC-MSMS in the 5 true positive patients so that the levels can be compared with false positive data in Appendix A. On line 269-270, the statement that accuracy and precision in this manuscript are comparable to other reports require appropriate referencing with values so that readers can make their own judgment  

Author Response

Response to Reviewer 2

Thank you for your review our paper on 17OHP by LCMSMS and it’s effect on screening performance of CAH in New Zealand. The responses to your comments are below

Thank you for pointing out our error in reporting that the number of positive screens are different in the text and in our tables. This has been corrected. I have also correct the PPV before and after LCMSMS implementation

Table 3 line 278

I accept the true positive results should be included in the appendix for comparison and have added them to the manuscript.

I have added the positive test results to the appendix for comparison.

I have referenced the appropriate papers to compare accuracy and precision to the method

Lines 289-293

“Our inter assay method precision range of 9.3-11.9% was comparable to other reports (7.9%-10.9%10, 3.9-18%18). The method is also sufficiently accurate, particularly when using the inter laboratory method mean as a target for certified CDC material. The method was linear beyond the clinical requirements for CAH investigations.”

Reviewer 3 Report

General

The authors present an LC-MS based 2nd tier method for newborn screening for CAH that is based on the determination of 17OHP. The 2nd tier method has been implemented in the newborn screening program and replaces a previously used 2nd tier method that was based on the same immunoassay used in the 1st tier bur with a more extensive sample pre-treatment that removes interfering compounds. The analytical performance characteristics of the method is described. Furthermore, the first two years of implementation of the new 2nd tier method is compared with the last two years of the original set-up. The primary findings are that the positive predictive (PPV) value of CAH screening was significantly increased, from 1.74% to 12.5%.

The study is well written and the finding are of interest to the newborn screening community, primarily due to the long term experience of implementation in routine newborn screening. On the other hand, the method itself is not novel and there are many studies that demonstrate the advantages of including several steroids in an LC-MS based second tier method for newborn screening. By adding more relevant steroids to the analysis it is likely possible to improve the PPV even more, as 12.5% is far from ideal. On balance, this demonstrates that LC-MS provides more accurate measurement as the technique is less susceptible to interferents commonly found in newborn screening samples. The authors demonstrate that this improved accuracy provides an substantial improvement in screening performance in routine screening.

The most important general issues that need to be addressed are:

Positive samples: What is the difference between samples that are above the 2nd tier cut-off and positive samples? Normally a sample that is above the cut-off is positive. This is important since there is a fundamental effect on how the PPV is calculated. Analytical set up: Why was Turboflow on-line solid phase extraction used? This is a fairly complex set-up and is not normally used in newborn screening. What are the pros and cons?

Detailed comments

2.2 Calibrators and Controls

- A serum substitute was used when producing calibrators and controls. Does this substitute contain SHBG? A large portion of 17OHP is bound to SHBG. This binding protein is an important factor when considering recovery.

- Later in the manuscript samples spiked with 17OH pregnalone is mentioned. The preparation of these samples should be described here.

2.3 Bloodspot Samples

- On what day of life are newborn screening samples taken in New Zealand? 17OHP levels change rapidly during the first days of life and are highly correlated to age at sampling. It is this important to know when the samples were taken and how much variation there is in sampling time.

- The CDC reference samples are later referred to as EQA samples. Please be consistent. Also, are these from the proficiency testing program or are they quality control materials? Only the quality control samples are certified.

2.5 Sample Preparation for the LCMSMS Method                                                                                            

The preparation of the internal standard solution needs to be described.

2.6 LCMSMS Analysis

This description in the first paragraph is a bit confusing. Please describe it as a Thermo TLX turboflow system set up for on-line solid phase extraction and then describe the various units in the system.

3.1 LCMSMS Method Performance

- Figure 1: Considering the number of samples run in the study it is surprising that it isn't possible to find an A and B sample that were run on the same column.

- Table 1: How were the assigned value determined? It is clearly not based on the enrichment values nor is it based on the mean values.

- Recovery: The recovery values are surprisingly good. If there was no SHBG present in the controls there is a clear risk of overestimating the extraction efficiency. It would be a good idea to investigate recovery using native blood.

3.2 Correlation between LCMSMS and Immunoassay

- In figure 4, the x-axis is labelled as log of mean concentration. Judging from that values, I believe that these are the concentration values, not the log.

4. Discussion

The authors suggest that including cortisol in the 2nd tier method is problematic due to the fact that premature infants are often treated with corticosteroids. This is a valid issue but is much broader that just affecting cortisol measurement. Treatment with cortisol and other corticosteroids suppresses the HPA axis, thus causing a suppression of 17OHP and most other steroids as well. Thus this not only an issue with regards to utilizing low cortisol as a biomarker. Please revise this part of the discussion.

The authors suggest that adjusting the sampling time would be an efficient way to optimize newborn screening for CAH. Please explain the reasoning. 17OHP levels are elevated at birth and drop during the first few days of life. For positive samples 17OHP continues to increase after with time. In many ways it is easier to screen for CAH with a later sample. If there is a wide range of sampling times it can be difficult to set an efficient cut-off. Judging from appendix A there is a wide range in sampling time. This can make it quite difficult to set a cut-off that provides a high PPV without compromising the methods sensitivity.

Author Response

Thank you for reviewing our manuscript.

Response to Reviewer 3

Thank you for your review our paper on 17OHP by LCMSMS and the effect on screening performance of CAH in New Zealand. The responses to your comments are below

“Positive samples: What is the difference between samples that are above the 2nd tier cut-off and positive samples? Normally a sample that is above the cut-off is positive. This is important since there is a fundamental effect on how the PPV is calculated.”

Response: We agree that this is an important point. Currently, there is no internationally agreed definition of a positive screening result and varying definitions occur depending on the screening jurisdictions. In New Zealand we define a positive newborn screening test as that which leads to a further unplanned action on a baby (Methods, lines 132-135). For example, although a low birth-weight baby may have an out-of-range 2nd tier result at 48 hours, this would not be considered a positive screen result if the recommended action is to await the next scheduled sample.  Conversely when an out-of-range result leads to an unplanned action on the baby (clinical referral or request of an additional unscheduled sample), this is considered to be a positive screen result. 

“Analytical set up: Why was Turboflow on-line solid phase extraction used? This is a fairly complex set-up and is not normally used in newborn screening. What are the pros and cons?”

Response: The technique was used as it prolongs the life of the analytical column and the technology was readily available on our sensitive LCMSMS system. Solvent extracted whole blood (dried) can result in protein and phospholipid particulates that can accumulate on the analytical column affecting performance and reduce column life. This explanation has been added to the re-submitted manuscript (lines 149-153)

“A combination of diffusion and size exclusion results in the retention of low molecular weight (<600 amu) polar and non-polar compounds. High molecular weight compounds are not retained and flow to waste. The method provides additional sample clean that removes protein particulates and phospholipids which can have a negative impact on analytical chromatography column performance and lifetime. “. We have also added details of the turboflow programming as an appendix

Detailed Comments

“A serum substitute was used when producing calibrators and controls. Does this substitute contain SHBG? A large portion of 17OHP is bound to SHBG. This binding protein is an important factor when considering recovery”

Response: We agree that protein binding is a key consideration and our serum substitute does not contain SHBG. 17OHP also circulates bound to CBG and albumin. All proteins are denatured and precipitated when the acetonitrile extraction solution is added to the bloodspots. Free polar steroids are then easily dissolved in the extraction fluid.

“Later in the manuscript samples spiked with 17OH pregnalone is mentioned. The preparation of these samples should be described here.”

We thank the reviewer for noting this and have added a description of  the preparation of 17-hydroxpregnenolone sulphate. The following has been added to the manuscript (lines 95-99)

“To evaluate the interference of sulphated steroids, 17-hydroxypregnenolone sulphate enriched samples were prepared by adding 240µL, 120µL, 60µL and 30µL of 333.3µM solution to 20mL of 55% haematocrit blood. Blood was mixed for 1 hour and spotted onto blood collection paper, dried at room temperature and stored at -20oC until analysis. The final concentrations of 17-hydroxypregnenolone sulphate were 500, 1000, 2000 and 4000nmol/L.”

“On what day of life are newborn screening samples taken in New Zealand? 17OHP levels change rapidly during the first days of life and are highly correlated to age at sampling. It is this important to know when the samples were taken and how much variation there is in sampling time.”

Response: We agree that sample collection time influences the normal range for 17OHP. As described in the manuscript (lines114-118) newborn screening samples in New Zealand are routinely collected between 48-72 hours of age. <1% of first samples are collected before 48 hours, however these are most often collected from sick and premature infants in whom further routine samples are expected and have little impact on our false positive rate. .

“The CDC reference samples are later referred to as EQA samples. Please be consistent. Also, are these from the proficiency testing program or are they quality control materials? Only the quality control samples are certified.”

Response: We thank the reviewer for noting this. The CDC samples used to assess analytical performance are certified quality control materials. The CDC material used to determine the correlation between LCMSMS and immunoassay were samples from a CDC newborn screening proficiency scheme. A further explanatory sentence has been added to the manuscript (lines 104-106 and 100-112)

“Certified quality control material (3 levels) enriched with known quantities of 17OHP were used to assess accuracy and recovery. Certified material was supplied by the Centre for Disease Control and Prevention (CDC, Atlanta, USA) Newborn Screening Quality Assurance Scheme.”

“33 residual external quality assurance samples supplied by a CDC Newborn Screening Proficiency Scheme were used to determine the correlation between LCMSMS and immunoassay methods of analysis.”

“The preparation of the internal standard solution needs to be described.

Response: Internal standard preparation is described in the calibrators and control section (Methods,lines 92-94).

“LCMSMS Analysis: This description in the first paragraph is a bit confusing. Please describe it as a Thermo TLX turboflow system set up for on-line solid phase extraction and then describe the various units in the system.”

Response: This description has been clarified, Methods, lines 144-165

“The LCMSMS system comprised an ultra high pressure liquid chromatography (UHPLC) quaternary pump combined with a TSQ Vantage mass spectrometer both from Thermo Fisher (Waltham, Mass., USA). A in-line Thermo Fisher turbulent flow (TLX) solid phase extraction was available for use. It comprised of a second UHPLC pump and A cyclone-P TurboflowTM (0.5 x 50mm) extraction column. TLX technology works as follows. Prepared samples are injected onto the turboflow column at high flow rate using the 1st UHPLC pump (Loading pump). A combination of diffusion and size exclusion results in the retention of low molecular weight (<600 amu) polar and non-polar compounds. High molecular weight compounds are not retained and flow to waste. The method provides additional sample clean that removes protein particulates and phospholipids which can have a negative impact on analytical chromatography column performance and lifetime.

Solvent A consisted of ultra-pure water (Milliq, Merck, Darmstat, Germany) with 0.05% formic acid and 2mM ammonium formate. Solvent B was 100% methanol. Solvent C was 100% acetonitrile. Solvent D was acetonitrile/isopropanol/acetone (45:45:10).

The turboflow UHPLC method contains a series of a steps that control pump flow rate, valve positions, step duration and mobile phase composition. Briefly, the prepared samples were injected onto the TFC column, via a 20µL sample loop, at a flow rate of 1.5mL/min (95% solvent A and 5% solvent C). Bound material was transferred to the analytical column by eluting the TFC column with 80% acetonitrile from an inline elution loop at a low flow rate (0.1mL/min) for 1 minute. Chromatographic separation was achieved by ramping of mobile phase B to 90% in 5 minutes before holding for 1 minute at 0.5 mL/min. During the chromatography phase the TLX column was washed with 100% mobile phase D before the elution loop was refilled with 80% acetonitrile. The total run time was 9.15 minutes. A detailed chromatography settings are shown in Appendix A.”

“Figure 1: Considering the number of samples run in the study it is surprising that it isn't possible to find an A and B sample that were run on the same column.”

Response: This is a valid point and has been addressed in the new manuscript (Figure 1).

“Table 1: How were the assigned values determined? It is clearly not based on the enrichment values nor is it based on the mean values.”

Response: The CDC provide a target value (their analytical data) plus a mean value of all particpating laboratories returning analytical data. We have discussed accuracy with respect to both of these values data. We have added this description to Table 1.

Recovery: The recovery values are surprisingly good. If there was no SHBG present in the controls there is a clear risk of overestimating the extraction efficiency. It would be a good idea to investigate recovery using native blood.”

Response: This is a valid point. We have added a further comment to the Discussion (lines  291-295).

“The mean recovery for three 17OHP concentrations was almost 100%, although a repeat recovery experiment in native blood to account for protein binding of 17OHP may be have been more appropriate. However, we assumed that protein precipitation would release bound 17OHP in the sample when the acetonitrile solution was added during the extraction phase of the procedure”

“In figure 4, the x-axis is labelled as log of mean concentration. Judging from that values, I believe that these are the concentration values, not the log.”

Response: We appreciate the reviewer noting this and have amended the label (Figure 4).

“The authors suggest that including cortisol in the 2nd tier method is problematic due to the fact that premature infants are often treated with corticosteroids. This is a valid issue but is much broader that just affecting cortisol measurement. Treatment with cortisol and other corticosteroids suppresses the HPA axis, thus causing a suppression of 17OHP and most other steroids as well. Thus this not only an issue with regards to utilizing low cortisol as a biomarker. Please revise this part of the discussion.”

Response: We agree that corticosteroid treatment is a wider issue which may reduce the reliability of any CAH screening biomarker.  We have removed our comment about corticosteroids and cortisol levels as we believe that a comprehensive discussion is beyond the scope of this paper. However, as a further piece of work we intend to evaluate additional second tier markers, and agree that the potential impact of corticosteroid treatment would be a key consideration here.  

“The authors suggest that adjusting the sampling time would be an efficient way to optimize newborn screening for CAH. Please explain the reasoning. 17OHP levels are elevated at birth and drop during the first few days of life. For positive samples 17OHP continues to increase after with time. In many ways it is easier to screen for CAH with a later sample. If there is a wide range of sampling times it can be difficult to set an efficient cut-off. Judging from appendix A there is a wide range in sampling time. This can make it quite difficult to set a cut-off that provides a high PPV without compromising the methods sensitivity.”

Response: We thank the reviewer for the opportunity to clarify this comment. The final routine sample is collected from extremely low birth weight babies at 4 weeks, at which point most have a corrected GA of  just 28-31 weeks.  Deferring collection of a final sample to unit discharge (typically at a corrected GA of ≥36/40) would allow for further adrenal maturation and would be anticipated to increase the specificity of a 17OHP screen. We have added an explanatory comment to the Discussion (lines 324-331).